# Generation of Primordial Germ Cell-like Cells from iPSCs Derived from Turner Syndrome Patients

**DOI:** 10.3390/cells10113099

**Published:** 2021-11-10

**Authors:** Aline Fernanda de Souza, Fabiana Fernandes Bressan, Naira Caroline Godoy Pieri, Ramon Cesar Botigelli, Tamas Revay, Simone Kashima Haddad, Dimas Tadeu Covas, Ester Silveira Ramos, Willian Allan King, Flavio Vieira Meirelles

**Affiliations:** 1Department of Veterinary Medicine, Faculty of Animal Science and Food Engineering, University of São Paulo (USP), Pirassununga 13635-000, Brazil; fabianabressan@usp.br (F.F.B.); nairagodoy@usp.br (N.C.G.P.); ramonbotigelli@gmail.com (R.C.B.); 2Department of Biomedical Sciences, Ontario Veterinary College (OVC), University of Guelph, Guelph, ON N1G 2W1, Canada; waking@ovc.uoguelph.ca; 3Department of Pharmacology, Institute of Biosciences (IBB), São Paulo State University (UNESP), Botucatu 18618-689, Brazil; 4Department Alberta Children’s Hospital Research Institute (ACHRI), University of Calgary, Calgary, AB T2N 4N1, Canada; tamas.revay@albertaprecisionlabs.ca; 5Center for Cell-Based Therapy, Regional Blood Center of Ribeirão Preto, Ribeirão Preto Medical School, University of São Paulo, Ribeirão Preto 14051-060, Brazil; skashimahaddad@gmail.com (S.K.H.); dimas@fmrp.usp.br (D.T.C.); 6Department of Genetics, Ribeirão Preto Medical School, University of São Paulo, Ribeirão Preto 14049-900, Brazil; esramos@fmrp.usp.br

**Keywords:** Turner syndrome, iPSCs, PGCLCs

## Abstract

Turner syndrome (TS) is a genetic disorder in females with X Chromosome monosomy associated with highly variable clinical features, including premature primary gonadal failure leading to ovarian dysfunction and infertility. The mechanism of development of primordial germ cells (PGCs) and their connection with ovarian failure in TS is poorly understood. An in vitro model of PGCs from TS would be beneficial for investigating genetic and epigenetic factors that influence germ cell specification. Here we investigated the potential of reprogramming peripheral mononuclear blood cells from TS women (PBMCs-TS) into iPSCs following in vitro differentiation in hPGCLCs. All hiPSCs-TS lines demonstrated pluripotency state and were capable of differentiation into three embryonic layers (ectoderm, endoderm, and mesoderm). The PGCLCs-TS recapitulated the initial germline development period regarding transcripts and protein marks, including the epigenetic profile. Overall, our results highlighted the feasibility of producing in vitro models to help the understanding of the mechanisms associated with germ cell formation in TS.

## 1. Introduction

In humans, Turner syndrome (TS) is caused by complete or partial monosomy of the X chromosome (45, X or mosaic 45, X and 46, XX) [1]. It is estimated that the incidence of TS, including partial or complete monosomy X is about 1 in 2000 live female births [2,3]. Individuals that present TS have highly variable phenotypes, such as short stature, bone abnormalities, heart disease and absence of secondary sexual characteristics caused by pre-pubertal ovarian failure [4,5,6]. Furthermore, it has been hypothesized that the pathogenesis of TS is also connected to the dysregulation of gene expression involving epigenetic modifications as DNA methylation [7,8,9]. Previous studies in mice with TS helped the understanding of the patterns of global genomic imbalance involved chromatin regulators and transcriptional regulators in humans [10]. However, the mice with TS, unlike humans, are typically fertile. Because of that, the murine model does not recapitulate the specification of germ cells and primary gonadal failure characteristics of TS patients. Since the human primordial germ cells (PGCs) from TS patients are challenging to obtain due to ethical procedures and scarce material, other in vitro models, such as the differentiation of induced pluripotent stem cells (iPSCs) into primordial germ cell-like cell (PGC-like cells or PGCLCs) can help understand the biological process involved in the pathogenesis of TS.

Somatic reprogramming through the overexpression of genes related to pluripotency led iPSCs to reprogramming [11,12]. iPSCs can self-renew indefinitely in in vitro culture while maintaining their capacity to differentiate into cells of the three embryonic germ layers [13]. Several strategies for generating iPSCs were developed by diverse groups combining the method by retroviruses (gamma retroviruses or lentivirus) and using non-integrative methods as Sendai virus and episomal plasmids [14,15,16,17,18,19]. Previous studies have demonstrated that the efficiency of episomal reprogramming could be increased by using peripheral mononuclear blood cells (PBMCs) enriched with a population of cells erythroblasts [20,21,22]. The use of PBMCs cells represents a much more affordable and abundant source of patient cells for reprogramming without extensive maintenance in culture [22,23].

In recent years, significant progress has been made in the reprogramming process of PBMCs into integration-free iPSCs to use as an experimental platform for in vitro disease models in humans to investigate pathophysiology and aid drug development [24,25,26,27,28,29]. However, at the moment, no studies in TS patients have been generated by reprogramming PBMCs into integrating-free iPSCs to study in vitro germ cells specification. The induction of iPSCs into primordial germ cells like cells (PGCs-like cells or PGCLCs) opened a new possibility to investigate the specification of germ cells, helping to understand the main events of their development [30,31]. Over the last few years, the field of PGCLCs is progressing so rapidly that recent studies have shown the possibility of inducing human PGCLCs in oogonia-like and pro-spermatogonia-like cells [32,33,34]; although the generation of in vitro germ cells with robust and long-term reconstitution potential still remains a significant challenge. In vitro germ cell-like cells (GCLCs) of TS have been reported via xenotransplantation into murine seminiferous tubules [35]. GCLCs-TS demonstrated differences in cell stage development and regulatory networks of transcription factors compared to PGCLCs. Another study showed the possibility of inducing iPSCs from TS in germ cells [36]. However, little information was described in this research regarding the methodology used to differentiate in PGCLCs and its subsequent characterization about gene expression and epigenetic profiles.

Herein we explored the use of non-integrtative vectors to reprogram adult blood cells from patients with Turner syndrome into iPSCs. In addition, we demonstrated the establishment of an efficient protocol for obtaining in vitro progenitor germ cells from these patients. The methodology developed in this study has potential applications in banks of iPSCs and PGCLCs. The in vitro modeling of germinative cells can help to understand the balance between transcribed genes and epigenetic profile associated with the pathogenesis of TS.

## 2. Materials and Methods

### 2.1. Ethics Statements

The experiment involved five TS patients and two healthy controls groups, and was reviewed and approved by National Research Ethics Commission of Brazil (number: 2.779.118) and by the Clinical Hospital of Ribeirao Preto from Ribeirão Preto Medical School/University of São Paulo (number: 81683317.1.1001.5440). The inclusion criteria of TS patients were age over 21 years, without intellectual disabilities, unrelated, with TS stigma and karyotype 45, X “pure”. The control group were composed by female and male aged over 21 years, without intellectual disability, unrelated, without TS stigma, whose karyotype contains only one cell line without numerical or structural aberrations in the chromosome. The human embryonic stem cell (hESCs) line H1 (cat.# WA01-cGMP Material) was bought from WiCell International Stem Cell Bank, and was first published by Thomson et al. [37].

### 2.2. PBMCs Expansion

PBMCs were collected and isolated following an adapted version of the protocol previously established [38]. The PBMCs were collected from a superficial vein of the arm thought Vacutainer tubes with Ficoll monolayer (cat.# 362761; BD Biosciences, San Diego, CA, USA), according to the manufacturer’s instructions. Approximately 1–2 × 10^6^ of PBMCs were cultured in StemSpanTM medium (cat.# 09655, Stem Cell Technologies, Vancouver, BC, Canada) supplemented with the following cytokines: 100 ng/mL Recombinant Human Stem Cell Factor (SCF) (cat.# 255-SC-01M, R&D SYSTEM), 10 ng/mL Recombinant Human IL-3 (cat.# 203-IL-010, R&D SYSTEM), 2 U/mL Recombinant Human Erythropoietin (EPO) (cat.# 2877-TC-500, R&D SYSTEM), 40 ng/mL Recombinant Human Insulin-like Growth Factor 1 (IGF-1) (cat.# 291-G1-200, R&D SYSTEM), and 1 μg/mL dexamethasone (cat.# D2915 Sigma-Aldrich Corp., St. Louis, MO, USA). The PBMCs were maintained and cultured at concentration of 2 × 10^6^ cell/mL 12 days at 37 °C, 5% CO_2_. Approximately at D12-D14 days the PBMCs enriched with erythroblastic population were used for reprogramming.

### 2.3. Feeder Cells

All experiments involving animals to produce MEFs were approved by the Animal Use Ethics Committee of the Ribeirão Preto Medical School, University of São Paulo (ethics committee number:039/2018). MEFs were isolated from 13.5d mouse fetuses (C57BL/6) and cultured on complete cultured in Iscove’s Modified Dulbecco’s Medium (cat.# 12200069, IMDM, Gibco Thermo Fisher Scientific) supplemented with 10% fetal bovine serum (FBS)(cat.# SH30088.03, GE Healthcare Life Sciences), 1% GlutaMAX (cat.# 35050061, Gibco Thermo Fisher Scientific), 1% MEM Non-Essential Amino Acids Solution (cat.# 11140050, Gibco Thermo Fisher Scientific) and 1% penicillin/streptomycin (cat.# 15070063, Gibco Thermo Fisher Scientific). Cells were treated with mitomycin C (cat.# M4287, Sigma Aldrich) and used as a feeder monolayer at 1 × 10^5^ cells in 6 well dishes. The dishes were treated with 0.1% gelatin previously to seeding.

### 2.4. Reprograming and Culture of hiPSCs

Reprogramming was performed as previously described before [39] with some modifications. Briefly, 2 × 10^6^ cell/mL of PBMCs cells (TS, n = 5; CTRL Female, n = 1; CTRL Male, n = 1) were washed in 0.5% Bovine Serum Albumin (BSA) (cat.# 15260-037, Gibco, Thermo Fisher Scientific, Waltham, MA, USA) diluted in 10 mL of Dulbecco’s phosphate-buffered saline (D-PBS) (cat.# 14190144—Gibco, Thermo Fisher Scientific, Waltham, MA, USA) and centrifuged at 200× *g* for 10 min at room temperature. The cellular pellets were homogenized with 82 μL nucleofector buffer and 18 μL of the supplement from Human CD34 Cell Nucleofector Kit (cat.# VPA1003—Lonza, Basel, Switzerland). The cells were mixed with episomal vectors: 2.1 μg pCE-h*OCT3/4* (cat.# 41813—Addgene), 2.1 μg pCE-*hSK* [*SOX2* and *KLF4*] (cat.# 41814—Addgene), 2.1 μg pCE-h*UL* [*L-MYC* and *LIN28*] (cat.# 41855—Addgene), 2.1 μg pCE-m*p53DD* [*p53 carboxy-terminal dominant-negative fragment*] (cat.# 41856—Addgene) and 1.6 μg pCXB-*EBNA1* (cat.# 41857—Addgene). The nucleofection was performed by electroporation using the Nucleofector 2b Device program T-16 (cat.# 1971C DNA—Lonza, Basel, Switzerland).

The electroporated cells were plated in 2 mL of PBMCs expansion medium supplemented with 20 ng/mL basic fibroblast growth factor (bFGF) (cat.# 233 FB-05, R&D SYSTEM) in one well of a 12 well plate.Two days after electroporation (D2) the cells were collected, centrifuged at 200× *g* for 5 min at room temperature, the pellet was diluted in 1 mL/well of mouse embryonic fibroblast (MEFs) medium, which consist of a DMEM High Glucose (cat.#11885084, Gibco, Thermo Fisher Scientific, Waltham, MA, USA) 10% FBS (cat.# SH30071.03, Global Life Sciences Solutions USA LLC, Marlborough, MA, USA) supplemented with com 20 ng/mL bFGF and 25 mmol/L sodium butyrate (NaB) (cat.# B5887-1g, Sigma-Aldrich Corp., St. Louis, MO, USA). The cells were plated in two wells of a 12 wells plate coated with feeder cells of MEFs mitotically inactivated. The plates were centrifuged at 200× *g* for 30 min for the adhesion of the cells on the MEFs and transferred to an incubator at 37 °C, 5% de CO_2_.

Three days later (D3), media of reprogramed cells were changed for hESCs medium consisting on in Knockout DMEM F12 (cat.# 12660012, Gibco, Thermo Fisher Scientific, Waltham, MA, USA), 10% Knockout Serum Replacement (KSR) (cat.# 10828028—Gibco, Thermo Fisher Scientific, Waltham, MA, USA), 0.1 mM GlutaMAX (cat.# 35050, Gibco, Thermo Fisher Scientific, Waltham, MA, USA), 0.1 mM MEM-non essential amino acid solution (cat.# 11140, Gibco), 0.1 mM beta-mercaptoethanol (β-Mercaptoetanol) (cat.# 21985, Gibco, Thermo Fisher Scientific, Waltham, MA, USA) supplemented with com 20 ng/mL bFGF and 25 mmol/L NaB. The hESCs medium was changed every day.

The first colonies were usually identified on D12–14. All lines of hiPSCs were expanded in hESCs media and MEFs until the third passage (p). To expand and start the characterization, we used three colonies from hiPSCs-TS of Pt.1, Pt.2 and Pt.3, and one colony of Pt.4 and Pt.5. Also, three colonies from hiPSCs of male and female control. For that, the colonies were picked manually under an inverted microscope after treatment with 0.5 mM EDTA (UltraPure™ 0.5M EDTA, pH 8.0 cat.# 15575-020, Gibco, Thermo Fisher Scientific, Waltham, MA, USA) diluted in D-PBS. After the third passage, the hiPSCs were maintained and expanded in mTeSR-1 (cat.# 85850, Stem Cell Technologies, Vancouver, BC, Canada) or E8 medium (Essential 8™ Medium cat.# A1517001—Gibco, Thermo Fisher Scientific, Waltham, MA, USA) on Geltrex (cat.# A14133-02, Gibco, Thermo Fisher Scientific, Waltham, MA, USA) according to the manufacturer’s instructions, and incubated at 37 °C, 5% CO_2_.

### 2.5. Differentiation of hiPSCs into Embryoid Bodies

In brief, two wells from a six-well plate were dissociated using 0.5 mM EDTA as described above. The hiPSCs-TS and controls lines were plated into one well Corning low attachment plates (cat.# 3471—Corning, New York, NY, USA) containing 3 mL of E8 medium supplemented with 2.5 mg/mL the polyvinyl alcohol (cat.# 341584-25G—Sigma-Aldrich Corp., St. Louis, MO, USA). The day after plating, the medium was changed to E6 medium (Essential E6 medium, cat.# A1516401, Gibco, Thermo Fisher Scientific, Waltham, MA, USA). hESCs-H1 (p30) were used as positive control of the differentiation. All EBs were incubated 37 °C, 5% CO_2_. The media were changed every three or four days. All EBs were collected at day 15 and RNA extracted for the transcript quantification assay.

### 2.6. Differentiation of hiPSCs into PGCLCs

All hiPSCs-TS lines of colony C1 (p20), hiPSCs female of colony C4 and male control of colony C1 (p20) and hESCs-H1 (p30) cultured in mTeSR-1 medium were first differentiated into EpiLC following the protocol by Hayashi et al. [30,40] with minor alterations. Approximately, 1 × 10^5^ cells were plated on Geltrex-coated 6 wells and cultured for 2 days in Epiblast medium, which consist in [48 mL de knockout DMEM F12 (cat.# 12660012, Gibco, Thermo Fisher Scientific, Waltham, MA, USA), 0.5 mL N2 supplement (cat.# 17502048, Gibco, Thermo Fisher Scientific, Waltham, MA, USA), 1 mL B2 supplement (cat.# A1486701 Gibco, Thermo Fisher Scientific, Waltham, MA, USA), 0.5 mL KSR (cat.# 10828028, Gibco, Thermo Fisher Scientific, Waltham, MA, USA), 20 ng/mL Activin A (cat.# 338-AC, R&D SYSTEM), and 12 ng/mL bFGF (R&D SYSTEM). At D2, epiblast-like cells (EpiLC) were passaged then induced into hPGCLCs following the protocol described by Irie et al., [41] with minor modifications. The EpiLC were plated above 2.5 × 10^6^ per well into 24 agree well low cell-binding V-bottom plate (cat.# 34415, Stem Cell Technologies, Vancouver, BC, Canada) with PGCLCs medium [for 10 mL: 7.853 mL GMEM (cat.# 11710035, Gibco, Thermo Fisher Scientific, Waltham, MA, USA), 15% KSR (Gibco), 0.1 mM (Gibco), 0.1 mM Sodium Piruvate (cat.#11360070, Gibco, Thermo Fisher Scientific, Waltham, MA, USA), 0.1 mM β-Mercaptoethanol (Gibco), 2 mM l-glutamine (Gibco), 10 μM Rock inhibitor (cat.# 1254, Tocris, Toronto, ON, Canada), 1.000 U/mL Human Recombinant Leucemia Inhibiory factor (cat.# 1010, ESGRO, ESG110, EMD Millipore Corporation, Billerica, MA, USA), 200 ng/mL de Human Recombinant Bone Morphogenetic protein 4 (BMP4) (cat.# 314-BP-010, R&D SYSTEM), 100 ng/mL SCF (cat. # 7466-SC, R&D SYSTEM), 50 ng/mL Human Recombinant Epidermal Growth Factor (EGF) (cat.# 236EG-200, R&D SYSTEM)]. The media was changed every other day (D4) and all procedure the cells were incubated 37 °C, 5% de CO_2_. hPGCLCs were collected at day six, for immunostaining and RNA extracted for the transcript quantification assay.

### 2.7. Screening for Vector Spontaneous Integration

Absence of vector integration was analysed by polymerase chain reaction (PCR) in hiPSCs-TS (p10) and controls (p10 and p20) using primers designed from sequences in literature [21] (Table 1). The DNA were extracted using a commercial kit (DNeasy Blood & Tissue Kit, cat.#69504—Qiagen, Frederick, MD, USA) according to the manufacturer’s protocol. For the amplification reaction, 0.1 µg of DNA of each hiPSCs lines was used, 10 pmols of each primer, 12.5 µL Mix Goldstar (cat.# Pk0064-2—Eurogentec, Maastricht, Netherlands) and 10.5 µL DNAse free (cat.# 10977015, Gibco, Thermo Fisher Scientific, Waltham, MA, USA) for a final volume of 25 µL. The reactions were performed in a thermocycler (MyCycler^®^ Thermal Cycler, Bio-Rad), and the PCR conditions consisted of an initial denaturation step at 94 °C for 10 min followed by 35 cycles of denaturation at 94 °C for 45 s, annealing from 57 °C for 45 s, elongation at 72 °C for 90 s and a final elongation step at 72 °C for 7 min. The PCR products were visualized on a 1.5% agarose gel (Invitrogen, Carlsbad, CA, USA) with SYBRsafe (cat.# S33102—Termo Fisher) after a run of 50 min at 110 V. Subsequently, the bands were analysed regarding the size of the fragment and the results were photo documented. For the positive controls 0.1 µg of each plasmid vectors were used.

### 2.8. Cytogenetic Analyses

The karyotyping was performed by Karyotekk Inc. at University of Guelph (Guelph, ON, Canada). For each hiPSC clone, a total of 20 metaphases were counted and eight analyzed; four metaphases were karyotyped at a band resolution of either 425–450 or 450–475.

### 2.9. Alkaline Phosphatase and Immunophenotypic Characterization

For the alkaline phosphatase (AP) detection, all lines and subclones of the hiPSCs-TS and controls lines (p1 and p10) and hESCs-H1 (p30) were washed three times with PBS and analyzed using a commercial kit (Leukocyte Alkaline Phosphatase Kit, Sigma, cat.# 86R, Sigma-Aldrich, St. Louis, MO, USA) according to the manufacturer’s protocol. 

For flow cytometry analyzes, approximately 1 × 10^6^ of hiPSCs-TS and controls lines (p20) and hESCs-H1 (p30) were dissociated as a single cell using Stem Pro-Accutase as described above. The cells were incubated with 1 mL of fixation/permeabilization Solution Kit with BD GolgiStop (cat.# 554715—BD Biosciences) for 10 min at room temperature, followed by washing with 1 mL PBS and centrifuged at 200× *g* for 3 min at room temperature. The cells were stained with monoclonal antibodies: Alexa Fluor^®^ 488 Mouse anti-Human Nanog (cat.# 560791—BD Biosciences), and PE Mouse anti-Oct3/4 (cat.# 560186—BD Biosciences) following the concentrations recommended by manufacturing. For surface marker, SSEA-4, the cells were washed in PBS as described above and incubated with monoclonal conjugated antibody Alexa Fluor^®^ 647 Mouse anti-SSEA-4 (cat.#MC813-70—BD Biosciences) following the concentrations recommended by manufacturing. In all reactions nonspecific immunoglobulin G of the corresponding class as PE Mouse IgG2a (cat.# 558595—BD Biosciences) and FITC Mouse IgG1 (cat.# 550616—BD Biosciences) served as the negative control. Afterwards, the cells were resuspended at 200 μL of PBS and analyzed on FACS Sort flow cytometer (Becton-Dickinson) using CellQuest software version 5.2.

For immunostaining, all lines of hiPSCs-TS and control lines (p20), hESCs-H1 (p30), and hPGCLCs-TS and controls lines (D6) were fixed in 4% paraformaldehyde solution (PFA) for 12 min, the cells were washed three times for 30 min in phosphate-buffered saline (PBS), and the cells were permeabilized with PBS and 0.1% Triton X-100 (TBST, Sigma) for 30 min. The cells were blocked with 1% bovine serum albumin (BSA, Sigma-Aldrich Corp., St. Louis, MO, USA) and 0.1% PBS Tween-20 (PBS-T) for one hour at room temperature. 

The primary antibodies were diluted in 0.01% BSA and 0.1% PBS-T, and included polyclonal anti-rabbit IgG anti-DDX4 (VASA) (1:500, ab13840, Abcam, Cambridge, England), polyclonal anti-rabbit IgG anti-DAZL (1:500, ab34139, Abcam, Cambridge, England), polyclonal anti-rabbit IgG anti-DPPA3 (STELLA) (1:500, sc67249, Santa Cruz Biotechnology, CA, USA), monoclonal anti-mouse IgG1 anti-TFAP2C (AP-2γ) (1:100, sc12762, Santa Cruz Biotechnology, TX, USA), polyclonal anti-goat IgG anti-SOX17 (1:400, AF1924; R&D Systems, Minneapolis, MN, USA), polyclonal anti-rabbit IgG anti- OCT4 (POU51F) (1:100, ab19857, Abcam, Cambridge, England), polyclonal anti-rabbit IgG anti-SOX2 (1:250, ab97959, Abcam, Cambridge, England), polyclonal anti-rabbit IgG anti-NANOG (1:500, ab21624, Abcam, Cambridge, England), polyclonal anti-rabbit IgG anti-KDM6A (1:200, NBP1 80628, Novus Biological, Danvers, MA, USA), polyclonal anti-rabbit IgG anti-H3K27me3 (1:500,7449, Millipore, Temecula, CA, USA), polyclonal anti-rabbit IgG anti-H3K9me2 (1:500, 7441, Millipore, Temecula, CA, USA), polyclonal anti-rabbit IgG anti-H4K20 (1:500, ab9051 Abcam, Cambridge, England), monoclonal anti-mouse IgG1 anti DNMT3B (1:50, sc-376043, Santa Cruz Biotechnology, CA, USA). The primary antibodies were incubated overnight at 4 °C temperature. Next day, the cells were washed two times with PBS for 30 min and one time with PBS-T for 30 min, next the cells were incubated with secondary antibodies diluted in 0.01% BSA and 0.1% PBS-T overnight at 4 °C temperature. The secondary antibodies included Alexa Fluor 488 donkey anti-rabbit IgG (1:500, A21206, Life Technologies, Waltham, MA, USA), Alexa Fluor 488 donkey anti-mouse (1:500, A21202, Life Technologies, Waltham, MA, USA) Alexa Fluor 594 donkey anti-goat IgG (1:500, A11058, Life Technologies, Waltham, MA, USA). 

The negative controls were obtained by omitting the primary antibodies. The cells were counterstained with a 1:1000 dilution of Hoechst dye (trihydrochloride, trihydrate, cat# 33342, Invitrogen, Carlsbad, CA, USA) and mounted with Prolong Gold antifade (cat# P36930, Life technology; Carlsbad, CA, USA). Fluorescence images for hiPSCs-TS and control lines were captured using an EVOS™ digital inverted microscope (Thermo Fisher Scientific, Waltham, MA, USA). hPGCLCs images were made on an inverted confocal Olympus (FV1200) microscope. 

### 2.10. Analysis of Gene Transcription Levels by RT-qPCR

hiPSCs-TS and control lines (p15 and p20), hESCs-H1 (p30), and hPGCLCs-TS and controls lines (D6) were submitted to total RNA extraction using the Trizol^®^ protocol (Thermo Fisher Scientific, Waltham, MA, USA). After extraction, total RNA samples were quantified with a NanoDrop^®^ (Thermo Fisher Scientific, Waltham, MA, USA). For DNA digestion and reverse transcription, we adjusted the concentration to 1.000 ng RNA per sample. All the samples were subjected to DNA digestion using a DNAse I—Amplification Grade^®^ (cat.# 18068015, Thermo Fisher Scientific, Waltham, MA, USA). For reverse transcription (RT) of the samples, a high-capacity cDNA reverse transcription kit was utilized (cat.# 4368814, Thermo Fisher Scientific, Waltham, MA, USA). 

Relative quantification of transcribed genes was analyzed by qPCR using the PowerUP SYBR Green^®^ PCR Master Mix reagent (cat.# A25742, Thermo Fisher Scientific, Waltham, MA, USA) in ABI-7500 real-time PCR system equipment. Then, qPCR reactions were run in a volume of 10 μL containing 100 nM of each primer, 1X Power UP SYBR Green, 2.5 μL H2O and 1 μL template (four-fold diluted cDNA; 25 ng). Cycling conditions for amplification were as follows: 95 °C for 15 min and 45 cycles at 95 °C for 15 s, 60 °C for 5 s, 72 °C for 30 s and finally 72 °C for 2 min. A melting curve analysis was performed to verify the amplification of the specific products, and all reactions were performed in triplicate. Additionally, ultrapure DNAse and RNAse free water was used as a negative control of the reaction.

The cycle threshold (Ct) values of the target genes were normalized to the Ct value of *beta actin* (*Β-ACTIN*) and *18S* housekeeping genes, and then, the relative gene expression was determined by 2^−∆Cq^ equation [42]. The primer sequences used for the X-linked signature (*XIST*, *ZFX*, *RNF12*, *MECP2* and *EZH2*) analyses have been previously described in the literature [28,43]. The primer sequences used for the germline (*SOX17*, *TFAP2C*, *PRDM14*, *NANOS 3*, *DPPA3*, *DAZL* and *VASA*), and epigenetic profile (*DNMT3A*, *DNMT3B*, *TET1*, *TET2* and *TET3*,) analyses have been previously described in the literature [41,44,45]. Other gene-specific primers for epigenetic profile as *DNMT1*, *IGF2* and *IGF2R,* also for pluripotency were designed using the Primer-BLAST (NCBI) software based upon sequences available in GenBank (Table 2). 

### 2.11. Statistical Analysis

The statistical analyses consisted on analysis of variance (two-away *ANOVA*, *p* < 0.05) followed by Tukey’s test to determine the differences in gene expression and the group means (*p* < 0.05) using R software [46] and GraphPad Prism 8 (GraphPad Software, San Diego, CA, USA). A Pearson correlational analysis (*p* < 0.05) was performed to verify the gene expression correlations among the different stages.

## 3. Results

### 3.1. Generation and Characterization of hiPSCs-TS

All PBMCs-TS of patient 1, 2 and 3 expanded in StemPan medium showing a cell growth similar to the controls lines. However, the PBMCs-TS of patient 4 and 5 demonstrated lower cellular expansion than those from controls lines and other patients (Appendix A). On day 12, all lines of PBMCs-TS and control cell lines enriched with erythroblastic cells were transfected with episomal vectors containing the following pluripotency genes: *OCT4*, *KLF4*, *SOX2*, *L-MYC* and *LIN28* and the other two, *p53* and *EBNA1*, to assist in the reprogramming process (Figure 1A) (Appendix A). The first small and compact hiPSCs-TS colonies of patients 1, 2 and 3 were observed approximately 12 days after transfection, similar to the control lines. On the other hand, the first colonies of hiPSCs-TS of patients 4 and 5 appeared approximately 20 days after cells transfection (Figure 1B). All hiPSCs-TS and control cell lines showed similar proliferation rates and colonies pattern (round and flat) as the human embryonic stem cells (hESCs) cultures. Three clones of hiPSCs control lines and hiPSCs-TS of patient 1, 2 and 3, and one clone of patients 4 and 5 were expanded in matrix and after successive passages were characterized for pluripotency and in vitro differentiation. The hiPSCs-TS from five patients (Pt.1, Pt.2, Pt.3, Pt.4, and Pt.5) were compared with two control cell lines for characterization analysis data. Also, the cell line hESCs-H1 was used as an internal control in all experiments.

The first pluripotency test performed was the alkaline phosphatase stain which showed that all hiPSCs-TS lines and controls had intense staining (Figure 1C) (Appendix A). Next, the presence of specific proteins, such as OCT4, SOX2, and NANOG were qualitative evaluated by immunostaining assay for (Figure 1C) (Appendix A). Both hiPSCs-TS and control lines (p20) showed intense staining for these three markers.

Gene expression analyses demonstrated abundance of *OCT4* (also known as *POU5F1*) and *NANOG*, mRNA in hiPSCs-TS cell lines expressed with similar level to the hiPSCs controls lines (*p* > 0.05) (Figure 1D). However, hiPSCs-TS of Pt.1 and Pt.2 demonstrated higher presence of *SOX2* transcript than hiPSCs controls (*p* < 0.05) (Figure 1D). In addition, hiPSCs-TS of Pt.5 showed a higher presence of *KLF4* transcript than other hiPSCs patients and controls lines (*p* < 0.0001). Also, *c-MYC* mRNA diverges among hiPSCs patients and controls lines (*p* < 0.05). Furthermore, we showed by PCR that the reprogramming vectors were eliminated from all lines of hiPSCs-TS (Appendix A).

Flow cytometry analyses demonstrated the presence of SSEA-4, NANOG, and OCT4 among hiPSCs-TS and control lines (p20) (Figure 2). hiPSCs-TS of Pt.1 and Pt.3 presented more than 70% positive cells for the SSEA-4 marker (Figure 2B). In addition, hiPSCs-TS of Pt.2, Pt.4 and Pt.5 showed about 61% of positive cells for SSEA-4 (Figure 2B). All hiPSCs-TS showed divergence in the presence of OCT4 and NANOG proteins. hiPSCs- TS of Pt.2 and Pt.5 showed the lower percentage of NANOG (around 28–21%) and for OCT4 (about 56–35%) compared to other patients and controls lines (*p* < 0.05) (Figure 2B). All hiPSCs control lines had more than 70% positive cells for NANOG, OCT4, and SSEA-4 markers (Figure 2B).

To assess the in vitro differentiation capacity of hiPSCs-TS and controls lines, the cells were cultured under specific conditions that permit spontaneous differentiation into EBs, which are three-dimensional structures that contain tissue from the three embryonic layers (ectoderm, endoderm and mesoderm). All hiPSCs-TS and control lines (including hESCs-H1) were able to differentiate into EBs (Figure 3A). Gene expression analyses demonstrated abundance of *RUNX1*, *CD34* and *GATA4* mRNA in all EBs-TS lines expressed with similar level to the EBs controls lines (*p* > 0.05) (Figure 3B). However, EBs-TS demonstrated lower presence of *NESTIN* transcript than EBs controls (*p* < 0.05) (Figure 3B). Also, *NCAM-1* mRNA diverges among EBs patients and male control line (*p* < 0.05). Furthermore, EBs-TS of Pt.4 showed a higher presence of *AFP* transcript than other hiPSCs patients and controls lines (*p* < 0.0001) (Figure 3B).

Cytogenetic analysis by GTG band was applied in all hiPSCs-TS and control to investigate the chromosomal stability. All hiPSCs control lines, and the hESCs-H1 did not show any identifiable abnormal karyotype (Figure 4A–C). hiPSCs-TS of Pt.1, 3, 4, and 5 (p20) exhibited 45, X karyotypes, characteristic normally associated with TS patients. No other structural or numeric modifications were detected (Figure 4D–G). In contrast, hiPSCs-TS of Pt.2 demonstrated 45, X, but also revealed reciprocal chromosome translocations between chromosomes 11 with breakage at the q arm, and chromosome 12 with a breakpoint at the p arm (der(11;12)(q10;10), der(11;12)(p10;p10)) (Figure 4H). This translocation was only present in hiPSCs-TS, since the somatic cells of Pt.2 showed only the chromosomal abnormality related to TS, 45, X constitutive karyotype (data not show).

### 3.2. Epigenetic Signature of hiPSCs-TS

Epigenetic variations play an important role in cellular reprogramming to the state of iPSCs. All hiPSCs-TS and control lines had the presence of the histone H3 lysine 27 trimethylation (H3K27me3), suggesting that the cells were in the primed state of pluripotency (Figure 5A). Moreover, the protein H3K27me3 was observed in the *Barr corpuscles* of the hiPSCs female control, a characteristic of inactive X chromosome (Xi) (Figure 5A). To confirm that hiPSCs female control presents Xi, we analyzed the X Inactive Specific Transcript (*XIST*) and X Active Specific Transcript (*XACT*) gene expression. The result suggests that the hiPSCs female controls had an active X chromosome and an inactive X chromosome (XaXi). Also, as expected, hiPSCs-TS did not present abundance of *XIST* mRNA, but showed transcripts of *XACT* (Figure 5B). Additionally, we analyzed the gene expression of X chromosome linked genes Enhancer of the Zeste Polycomb 2 (*EZH2*) and the Zinc Finger Protein (*ZFX*) due to their role in cellular self-renew and acquisition of pluripotency (Figure 5B). hiPSCs-TS had similar abundance of *EZH2* and ZFX mRNA compare to hiPSCs control lines (*p* > 0.05).

### 3.3. Induced and Characterization of hPGCLCs-TS

The hiPSCs-TS from four patients (Pt.1, Pt.2, Pt.3 and Pt.4) and two control cell lines were cultured in mTeSR-1 medium (without 4i supplementation) were able to produce EpiLC, followed by a second step of differentiation into hPGCLCs (Figure 6A–F) (Appendix A). Also, the cell line ESCs-H1 was induced into hPGCLCs to be used as an internal control in all experiments.

Gene expression analysis showed reduction of *OCT4*, *NANOG* and *SOX2* mRNA abundance in the hiPSCs-TS stage towards hPGCLCs-TS (*p* < 0.05) (Figure 6B) (Appendix A). These results are in agreement with our immunoassay analysis, where the staining for SOX2 and NANOG proteins were not observed in both hPGCLCs-TS and control cell lines (data not show), suggesting that these cells reached the germ cell specification stage and pluripotent factors were no longer essential. Furthermore, we demonstrated the presence of early germ cell signature in both hPGCLCs-TS and control cell lines by the presence of abundance of *TFAP2C* mRNA and SOX17 and AP-2γ (also known as TFAP2C) proteins, suggesting the importance of both markers at the induction of hPGCLCs (Figure 6A–F) (Appendix A). Surprisingly, hPGCLCs-TS and control cell lines showed a low presence of *PRDM14*, *NANOS3*, *DPPA3* (also known as *STELLA*) and *VASA* (also known as *DDX4*—DEAD [Asp-Glu-Ala-Asp] box polypeptide 4) mRNAs (Figure 6B) (Appendix A). Moreover, *DAZL* mRNA (also known as deleted in azoospermia like) associated with late germ cells was not present in our hPGCLCs-TS and control cell lines (Figure 6B) (Appendix A). In agreement with our immunoassay analysis, we observed few hPGCLCs-TS positive for DAZL and DDX4 (Figure 6A–F) (Appendix A). Interestingly, our results demonstrated that the *PRDM14* and *DAZL* mRNAs were present in hiPSCs-TS and control lines stage towards EpiLC, suggesting that both genes display an importance to maintain the pluripotency stage.

The epigenetic profile of hPGCLCs-TS and controls lines showed a high presence of H3K27me3 protein when compared to the histone H3 lysine 9 dimethylation (H3K9me2) (*p* < 0.05) (Figure 6C–F) (Appendix A). Lysine-specific demethylase 6a (KDM6A, also known as UTX), H4K20 and DNMT3B proteins were present in hPGCLCs-TS and control cell lines (Figure 6C–F) (Appendix A).

Furthermore, gene expression analysis showed a high presence of *DNMT1* whereas *EZH2* mRNA were higher in the hPGCLCs-TS compared to hiPSCs-TS and hEpiLC stages (*p* < 0.05) (Figure 7A). Interestingly, hPGCLCs-TS did not demonstrate difference in the presence of *DNMT3A*, *DNMT3B*, *TET1*, *TET2*, *TET3* mRNA from the hiPSCs-TS stage towards hPGCLCs-TS. In addition, we also demonstrate the epigenetic profile of hPGCLCs control lines (Appendix A).

The Pearson correlation results demonstrated that each stage of induction of hPGCLCs had its own epigenetic profile. In hiPSCs-TS *DNMT1* was positively associated with *IGF2R* gene expression (0.97). Furthermore, *DNMT3A* was positively associated with *IGF2* (0.85), and *TET3* (0.98) mRNA amount. Moreover, *DNMT3B* was negatively associated with *DNMT1* (−0.75), *IGF2* (−0.37), *IGF2R* (−0.87), *TET2* (−0.54), *TET3* (−010), *XIST* (−0.43), and *XACT* (−0.41) (Figure 7B).

In hEpiLC-TS stages presented that *DNMT1* mRNA was positively associated with *IGF2* (0.57), *IGF2R* (0.73), *TET1* (0.22), *TET2* (0.82), *TET3* (0.90), *XACT* (0.75), and EZH2 (0.99) expression. In addition, *DNMT3A* and *B* was negatively *DNMT1*, *IGF2*, *IGF2R*, *TET1*, *TET2*, *TET3*, *XIST*, *XACT*, and EZH2 expression.

In hPGCLCs-TS stage demonstrated that *DNMT1* gene expression was positively associated (>0.90) with *TET1*, *TET2*, *TET3*, and *IGF2R*. Also, the *DNMT3A* was positively associated (0.80) with *DNMT3B* at hPGCLCs-TS and hEpiLC-TS stages (Figure 4B). However, *DNMT3B* was negatively associated with *IGF2* (−0.83), *EZH2* (−0.76), and *XACT* (−0.40) expression. Interestingly, *DNMT3B* was positively associated with *IGF2R* (0.42) and *TET1* (0.45) (Figure 7B).

## 4. Discussion

Advances in vitro cell reprogramming and differentiation have significantly contributed to our understanding of the biological processes of cell development. In particular, the creation of research models using iPSCs is an intriguing approach for the study of molecular and cellular alterations in initial formation of germ cells from aneuploid syndrome such as TS. Our study demonstrated a robust and efficient method to reprogramming PBMCs-TS into iPSCs, and their differentiation into PGCLCs. Our findings are in agreement with previous reports, which demonstrated that expansion and culture of PBMCs in a specific medium with a combination of interleukins to deplete the lymphoid population and expand the erythroblast/erythroid population, drives the cell cycle of proliferation, as well as gives these cells a unique epigenetic signature that facilitates the reprogramming process [22,39]. 

The potential for pluripotency in iPSCs cells is usually assessed primarily through the presence of the classic pluripotent transcription factors, such as *NANOG* (Nanog homeobox), *OCT4* (Octamer-binding transcription factor 4), and *SOX2* (SRY-related high-mobility group (HMG)-box-2). These three factors interact to keep their expression levels regulated, being able to activate genes involved in the maintenance of pluripotency, and to repress the expression of genes related to differentiation [47]. Our study showed that all hiPSCs-TS and control lines maintained the undifferentiated state and pluripotency shown by a robust expression of pluripotent transcripts and proteins comparable to ESCs. Also, hiPSCs-TS demonstrated the capacity of in vitro differentiation into EBs, and showed the presence of genes related to three embryonic layers corroborating with previous studies of hiPSCs from TS [27,28,29]. Furthermore, our results demonstrate that all hiPSCs-TS and control lines presented *EZH2* and *ZFX* transcripts, indicating self-renewal ability, as also described previously [48,49].

Chromosomal instability in iPSCs was observed after long-term culture. Culture conditions, such as culture medium and/or cytokines, cell passage methodologies, and age of cell donors, also influenced the presence of chromosomal abnormalities. Generally, ESCs and iPSCs have a progressive tendency to acquire extra chromosomes, affecting more frequently chromosomes 1, 12, 17, and 20 [50]. Also, the gain of chromosome 12 and trisomy of chromosome 8 seem to be common in hiPSCs cells [51]. Our results identified derivative chromosome translocation between the short and long arms of chromosomes 11 and 12 in the hiPSCs-TS of Pt.2. We believe these abnormal chromosomes could be related to the rapid de-differentiation process in hiPSCs-ST of Pt.2 and may relate to the low level of *NANOG* protein and transcript. Chromosomal abnormalities in sex chromosomes, such as additional copies of the X chromosome, have also been observed in pluripotent cells by different research groups [52,53]. In particular, a previous study with TS- hiPSCs reported restoring the chromosomal composition of cells with a 45,X karyotype to a 46,XX karyotype through the compensatory uniparental disomic process [28]. In our results, the hiPSCs-TS of five patients and subclones did not show any changes in the chromosomal composition in sex chromosomes, remaining with the 45, X karyotype unaltered.

The central strategy for generating PGCLCs in humans lies in the pluripotency stage of iPSCs that can be obtained in culture through the addition of small molecules (e.g., known as 4i: mitogen-activated protein kinase (MEK), glycogen kinase synthesis 3 (GSK3), p38 and JNK inhibitors) [41,54]. Then, human iPSCs will be differentiated into post-implantation human epiblast-like cells (hEpiLC) or mesoderm-like cells (hiMeLCs); this step corresponds to the beginning of human embryo formation, in which germ cells will emerge. Thus, hEpiLC and hiMeLCs exposed to the BMP4 signal and supporting cytokines will be transformed into the state of PGCLCs, which can resemble PGCs in vivo seven weeks after conception. Previous studies of human germline trajectory revealed that the PGCs and PGCLCs specification could be induced from a transient stage of naive-primed pluripotent stem cells [55,56]. Our results suggest that hiPSCs-TS and controls lines were induced in PGCLCs with an initial germinal signature, as they have TFAP2C and SOX17 markers. As observed in other studies [41,54,57] repressing somatic cells and inducing the germline specification is promoted by the presence of the *TFAP2C* gene, which regulates the *SOX17* gene and stimulates the BLIMP1 transcripts [58]. 

Post-translational modifications of histones influence the chromatin structure and play a role in regulating gene transcription and genome integrity [59]. Our immunofluorescence assay analyses showed higher levels of histone H3K27me3 in all hPGCLCs -TS and control lines, whereas the histone H3K9me2 showed low intensity. These results are in agreement with previous studies performed in PGCs in vivo [60,61]. Our results of the gene expression analyze showed that the differentiation of hiPSCs-TS into hPGCLCs increased the presence of *EZH2* mRNA, which catalyzes the addition of methyl groups to histone H3K27me3. It is well known that *KDM6A* is a gene that is present on the X chromosome, which contains the Jumonji domain 3 (Jmjd3) family that mediates histone H3K27me2 and H3K27me3 [62]. The KDM6A, through demethylation activity, influences cell differentiation and development [63]. Since *KDM6A* is one of the genes that escape XCI, it was thought to be associated with TS phenotype perhaps by regulating germ cell differentiation [7]. Nagaoka and collaborators [64] demonstrated that mouse PGCs deficient in the *kdm6 a/b* gene showed aberrant epigenetic reprogramming, which they are transmitted during embryo development in vivo. In our results, KDM6A protein was present in hPGCLCs-TS and control lines, suggesting that the presence of this protein may influence the promotion and proliferation of germ cells. However, further experiments, such as interfering in the *KDM6A* gene in hPGCLCs, are necessary to confirm if this gene’s influence as a critical regulator during germ cell induction. 

The global erasure of DNA methylation in hPGCs-like cells begins during migration (approximately 5 weeks of gestation), colonization, and sexual differentiation of germ cells. This erasure event is important for deleting imprinting genes and the reactivation of the X chromosome in women [65]. As demonstrated herein, the generated hPGCLCs-TS lines were in the early stage of germ cell specification, so the cells have not yet gone through DNA demethylation. However, hPGCLCs-TS may undergo DNA methylation as they progress through cell development. Previous studies have reported that hPGCLCs had limited epigenetic reprogramming, which differentiates them from in vivo PGCs [66]. In this regard, it will be interesting to further investigate the global DNA methylation after a long-term culture of hPGCLCs-TS.

## 5. Conclusions

In summary, our results showed that hiPSCs-TS had a pluripotency competency profile leading to cellular self-renewal, thus allowing differentiation into hPGCLCs-TS. The hPGCLCs-TS retains an initial germ cell identity and recapitulates the histone epigenetic profile similar to PGCs in vivo during the specification development stage. This approach allowed the study of the potential of an in vitro model in TS patients, which could assist the investigation of tissue differentiation and the potential genetic and epigenetic mechanisms that result in infertility. Finally, our work leads to new questions for future experiments as the differentiation of hPGCLCs from patients with TS in gametes, which could allow a better understanding of the cellular and molecular mechanisms that control the development of germ cells.

## Figures and Tables

**Figure 1 cells-10-03099-f001:**
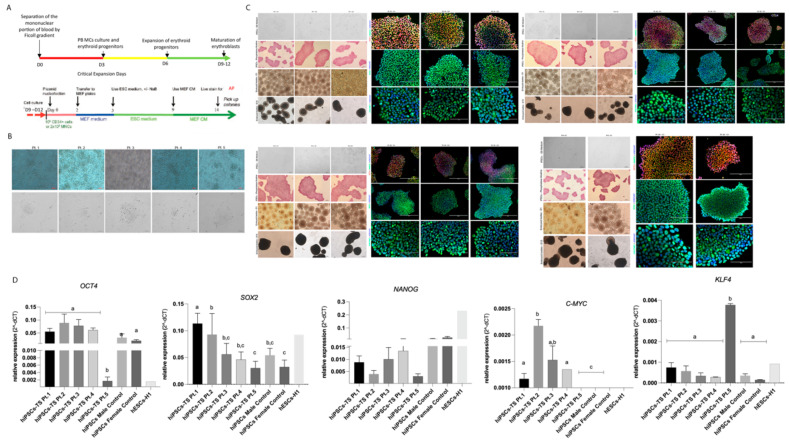
The process of reprogramming of Turner syndrome PBMCs into hiPSCs and characterization of hiPSCs-TS and colonies. (**A**) Schematic protocol used to reprogramming PBMCs into hiPSCs. (**B**) PBMCs with ten days of expansion in a specific medium for enrichment erythroblastic populations; the emergence of the first hiPSCs-TS colonies maintained in hESCs medium on MEF. (**C**) Characterization of hiPSCs-TS of Pt.1, Pt.2, Pt.3, Pt.4 and Pt.5. Phase contrast image showed hiPSCs maintained in mTeSR-1 medium on Geltrex; positively stain of alkaline phosphatase; Immunofluorescence assay showed the presence of pluripotency markers OCT4, SOX2, NANOG. Nuclei were stained with Hoechst (blue). Scale bars s: 200 μm and 100 μm. The data was compiled from at least three technical replicates. (**D**) Quantification of the relative expression of *OCT4*, *SOX2*, *NANOG*, *KLF4*, and *c-MYC* genes in all hiPSCs lines and subclones (p15). Individual quantifications by qRT-PCR were normalized using the *Β-ACTIN* gene. The ESCs-H1 line was used as a positive control and was not considered in the statistical model. All qRT-PCR from hiPSCs-TS and control data were compiled from at least three biological and technical replicates. *p* values were calculated by Two-way ANOVA, followed by Tukey’s multiple comparisons test as appropriate (letters a-b-c show difference between groups). Erros bars denote represent SD.

**Figure 2 cells-10-03099-f002:**
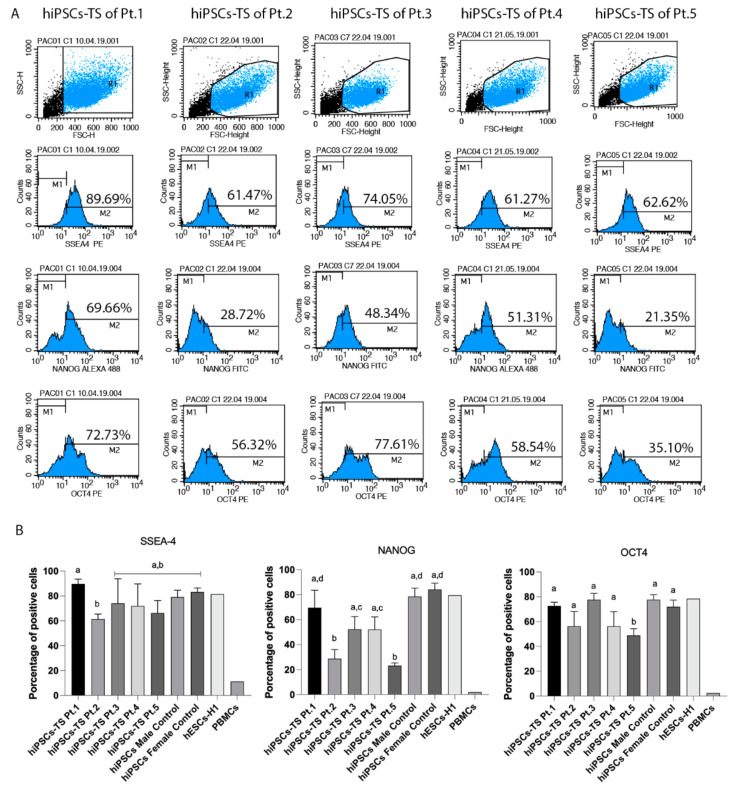
Flow cytometry analysis demonstrated the presence of pluripotent markers (SSEA-4, NANOG and OCT4) among all lines of hiPSCs-TS. (**A**) Histograms analysis showed in M2 the gate of positive cells present in each marker and cell line. (**B**) Percentage of positive cells in each line of hiPSCs-TS and hiPSCs controls lines. *p* values were calculated by Two-way ANOVA, followed by Tukey’s multiple comparisons test as appropriate (letters a-b-c-d show difference between groups). Error bars denote represent SEM. All flow cytometry analyses from hiPSCs-TS of Pt.1, Pt.2 and Pt.3 and control data were compiled from at least three biological and technical replicates. The data of hiPSCs-TS of Pt.4 and Pt.5 were compiled from three technical replicates.

**Figure 3 cells-10-03099-f003:**
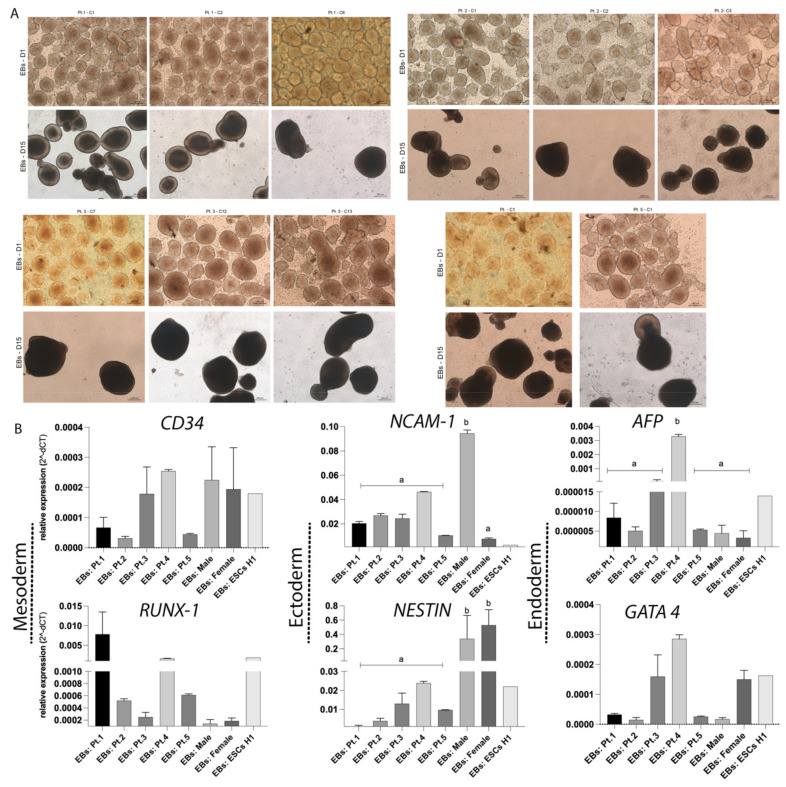
The in vitro differentiation capacity of hiPSCs-TS and subclones into EBs. (**A**) Embryoid body formation after 24 h and 15 days of differentiation. Scale bar: 200 μm. (**B**) Quantification of the relative expression of *NESTIN*, *NCAM1*, *RUNX1*, *CD34, AFP* and *GATA-4* genes in all EBs lines and subclones (D.15). Individual reactions of qRT-PCR were normalized to the *Β-ACTIN* gene. The ESCs-H1 line was used as a positive control and was not considered in the statistical model. All qRT-PCR from EBs-TS and control data were compiled from at least three biological and technical replicates. *p* values were calculated by Two-way ANOVA, followed by Tukey’s multiple comparisons test as appropriate (letters a–b show difference between groups). Erros bars denote represent SD.

**Figure 4 cells-10-03099-f004:**
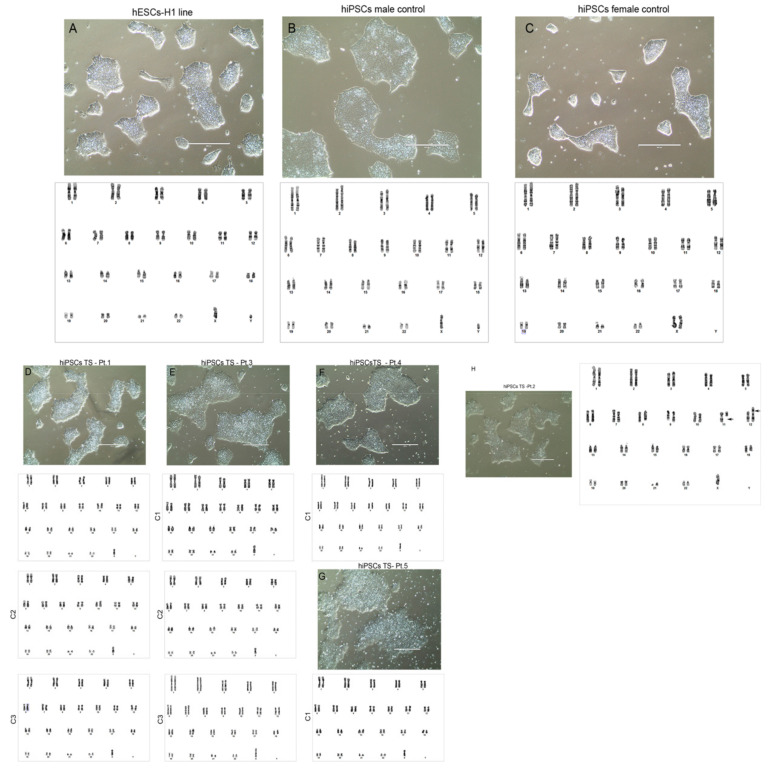
Cytogenetic analysis of all lines of hiPSCs-TS and controls. (**A**–**C**) phase contrast and karyotype of hESCs-H1 (p30) and hiPSCs male and female control (p20). 2). (**D**–**G**) phase contrast and karyotype of hiPSCs TS (p20). 3). (**H**) Phase contrast and karyotype of hiPSCs TS–Pt.2 (p20). Cytogenetic analyses revealed reciprocal chromosome translocations between chromosomes 11 with breakage at the q arm, and chromosome 12 with a breakpoint at the p arm (der(11;12)(q10;10), der(11;12)(p10;p10)). Scale bars indicate 400 μm.

**Figure 5 cells-10-03099-f005:**
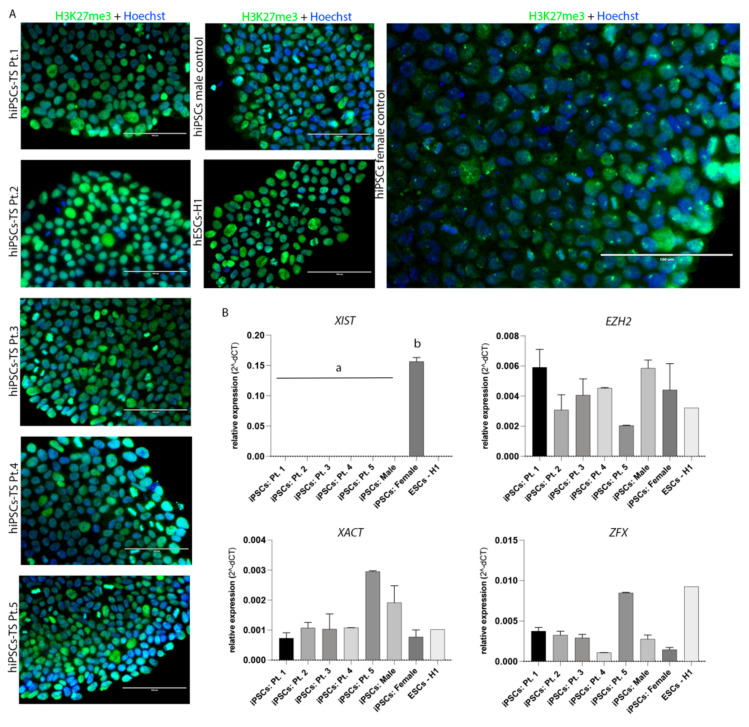
Epigenetic signature. (**A**) Immunofluorescence assay showed the presence of histone H3K27me3 in all lines of hiPSCs-TS and controls. At hiPSCs female control was observed the staining for H3K27me3 in small dots, the *Barr corpuscles*. Nuclei were stained with Hoechst (blue). Scale bars: 200 μm and 100 μm. The data was compiled from at least three technical replicates. (**B**) Quantification of the relative expression of *XIST*, *XACT*, *EZH2*, and *ZFX* genes in all hiPSCs lines and colonies (p15). Individual reactions of qRT-PCR were normalized using the *Β-ACTIN* gene. The ESCs-H1 line was used as a positive control and was not considered in the statistical model. All qRT-PCR from hiPSCs-TS and control data were compiled from at least three biological and technical replicates. *p* values were calculated by Two-way ANOVA, followed by Tukey’s multiple comparisons test as appropriate (letters a-b show difference between groups). Erros bars denote represent SD.

**Figure 6 cells-10-03099-f006:**
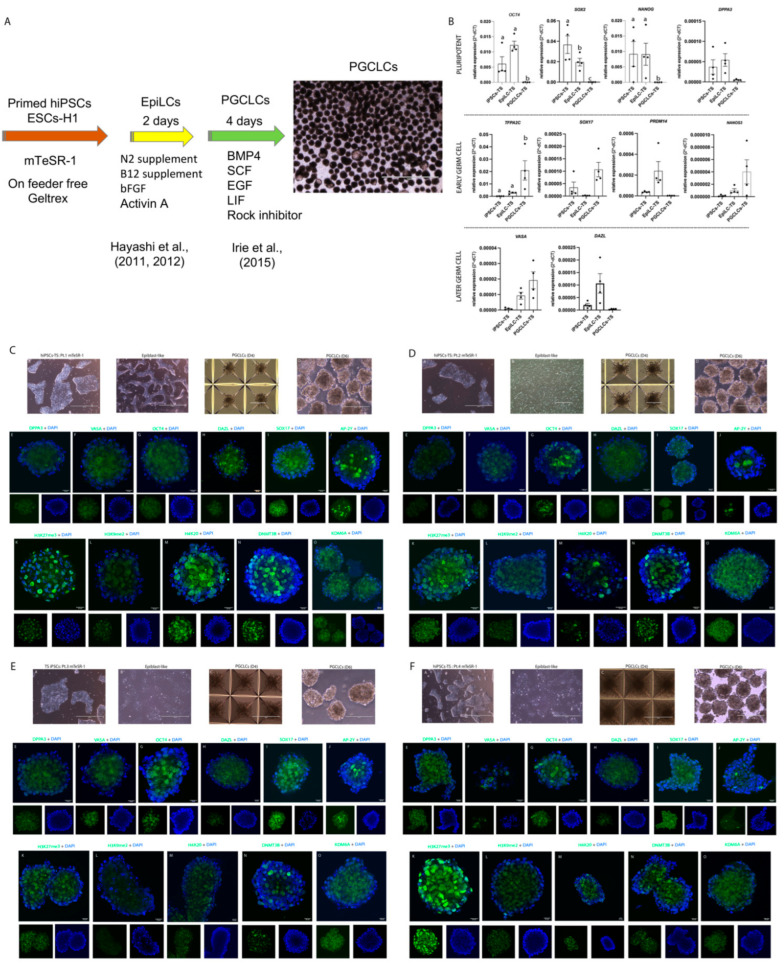
Dynamics of pluripotent and germ cells markers during hPGCLCs -TS induction. (**A**) Schematic protocol used to generate hPGCLCs. (**B**) hiPSCs-TS (p20), hEpiLC-TS (D.2), and hPGCLCs-TS (D.6) quantification of the relative expression of *OCT4, SOX2, NANOG, DPPA3, TFAP2C, SOX17, PRDM14, NANOS3, VASA,* and *DAZL* genes associated with germ cells development. Individual reactions of qRT-PCR were normalized to the *Β-ACTIN* gene. The data was compiled from at least four biological replicates and three technical replicates. *p* values were calculated by Two-way ANOVA, followed by Tukey’s multiple comparisons test as appropriate. (letters a-b-c show difference between groups). Error bars represent SEM. (**C**–**F**) PGCLCs-TS of Pt.1; PGCLCs-TS of Pt.2; PGCLCs-TS of Pt.3; PGCLCs-TS of Pt.4 showed. Phase contrast image showed hiPSCs-TS maintained in mTeSR-1 medium on Geltrex; hiPSCs-TS(p20) stimulated via hEpiLC; PGCLCs -TS (d4) cultured in Agreewell plates. Scale bars: 200 μm; hPGCLCs-TS (D.6) isolated for further characterization. Scale bars: 400 μm. Immunofluorescence analysis showed pluripotency and germ cell markers DDPA3, VASA, OCT4, DAZL, SOX17, and AP-2γ. Nuclei were stained with Hoechst (blue). Scale bars: 20 μm. Immunofluorescence analyzed showing the epigenetic profile of histones H3K27me3, H3K9me2, and H4K20; and DNMT3B and KDM6A protein markers. Nuclei were stained with Hoechst (blue). Scale bars: 20 μm. The data was compiled from at least two technical replicates for all staining.

**Figure 7 cells-10-03099-f007:**
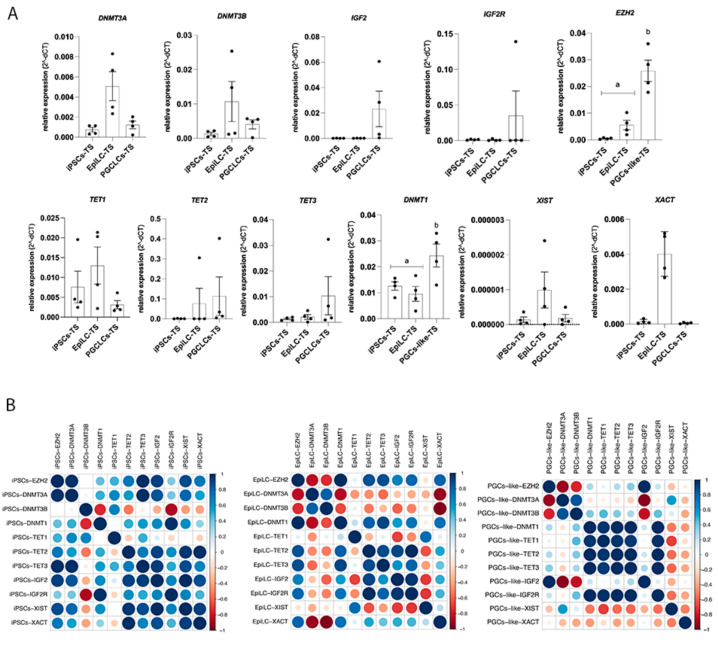
Epigenetic profile during PGCLCS-TS induction. (**A**) hiPSCs-TS (p20), hEpiLC-TS (D.2), and hPGCLCs-TS (D.6) quantification of the relative expression of genes associated with epigenetic marks. Individual reactions of qRT-PCR were normalized to the *Β-ACTIN* gene. The data was compiled from at least four biological replicates and three technical replicates. *p* values were calculated by Two-way ANOVA, followed by Tukey’s multiple comparisons test as appropriate. (letters a-b show difference between groups). *EZH2* gene (*p* < 0.0001); *DNMT1* gene (*p* < 0.005). Error bars represent SEM. (**B**) Gene expression correlations were determined according to the *Pearson* coefficient. The correlation (*Pearson*) profiles of the epigenetics genes during the hiPSCs-TS, hEpiLC-TS, and hPGCs-TS periods. Dark blue indicates a positive correlation, and dark red indicates a negative correlation.

**Table 1 cells-10-03099-t001:** List of primer for screening spontaneous integrations of episomal plasmid vectors.

Primer	Forward Primer Sequence	Reverse Primer Sequence
*EBNA-1*	ACGATGCTTTCCAAACCACC	CATCATCATCCGGGTCTCCA

**Table 2 cells-10-03099-t002:** Sequence of RT-qPCR primers related to experimental procedures.

Gene Name	Forward Primer Sequence	Reverse Primer Sequence	Annealing (°C)
*18S*	GCGAGTACTCAACACCAACATCG	TCAAGTCTCCCCAGCCTTGC	55
*Β-ACTIN*	TGAAGTGTGACGTGGACATC	GGAGGAGCAATGATCTTGAT	60
*OCT4 (POU5F1)*	CCTCACTTCACTGCACTGTA	CAGGTTTTCTTTCCCTAGCT	55
*SOX2*	CCCAGCAGACTTCACATGT	CCTCCCATTTCCCTCGTTTT	55
*KLF4*	GATGAACTGACCAGGCACTA	GTGGGTCATATCCACTGTCT	55
*MYC*	TGCCTCAAATTGGACTTTGG	GATTGAAATTCTGTGTAACTGC	55
*NANOG*	TGAACCTCAGCTACAAACAG	TGGTGGTAGGAAGAGTAAAG	55
*RUNX1*	CCCTAGGGGATGTTCCAGAT	TGAAGCTTTTCCCTCTTCCA	60
*CD34*	TGAAGCCTAGCCTGTCACCT	CGCACAGCTGGAGGTCTTAT	60
*AFP*	AGCTTGGTGGTGGATGAAAC	CCCTCTTCAGCAAAGCAGAC	60
*NCAM*	ATGGAAACTCTATTAAAGTGAACCTG	TAGACCTCATACTCAGCATTCCAGT	60
*NESTIN*	GCGTTGGAACAGAGGTTGGA	TGGGAGCAAAGATCCAAGAC	60
*XIST*	CGGTACGTTGAAGTTAGGGAATG	GTGCTGTCTAATCCAATGGGTAG	60
*XACT*	TGAATGAAGGCACATCTGA	CTCTCCCAGCCTATTTGTGG	60
*ZFX*	ATGGAAGAAGCAGATGTGTC	TTGAGGCTGAAGTAATGTCA	60
*RNF12*	ACCGATTGGATCGAGAAGAAGC	TGTAGTCGTCTCAGCAACTCT	60
*EZH2*	ACCGGTTGTGGGCTGCACAC	TGCAGCGGCATCCCGGAAAG	60
*MECP2*	CCAGGACTTGAGCAGCAGCG	CGGGAAGCTTTGTCAGAGCCC	60
*SOX17*	GAGCCAAGGGCGAGTCCCGTA	CCTTCCACGACTTGCCCAGCAT	65
*TFAP2C* *(AP-2γ)*	CGCTCATGTGACTCTCCTGACATCC	TGGGCCGCCAATAGCATGTTCT	60
*PRDM14*	CTACCGAGCCCGAGTGGCCTAC	TAGAGCCATCCCGGGACCGCA	60
*NANOS3*	CCCGAAACTCGGCAGGCAAGA	AAGGCTCAGACTTCCCGGCAC	60
*DPPA3* *(STELLA)*	ACGCCGATGGACCCATCACAGTTT	TCTCGGAGGAGATTTGAGAGGCCC	60
*DAZL*	TGGCCCTTCTTTCAGTGACTTC	GACCCTAGGGGGCACTAGTAA	60
*DDX4* *(VASA)*	TTCTTCACAAGCTCCCAATCCA	TTCTTCTCTGCATCAAAACCACA	60
*DNMT1*	GGGCTACCTGGCTAAAGTCAA	CTGCCATTCCCACTCTACGG	60
*DNMT3A*	TGGGATTCATCCAGACTCATGC	AAAGTGAGAAACTGGGCCTGAA	60
*DNMT3B*	TAACTGGAGCCACGACGTAAC	GCATCCGTCATCTTTCAGCCTA	60
*TET1*	GCTGCTGTCAGGGAAATCAT	ACCATCACAGCAGTTGGACA	60
*TET2*	CCAATAGGACATGATCCAGG	TCTGGATGAGCTCTCTCAGG	60
*TET3*	TCGGAGACACCCTCTACCAG	CTTGCAGCCGTTGAAGTACA	60
*IGF2*	GGGAGTTCTGGGGTAGGAAG	GAAAAATGCCCCAAGAAACA	60
*IGF2R*	CTTGACTGGGGCAATGATTT	GACGCATCGGGTGTAACTTT	60

## Data Availability

The authors confirm that all data and materials support the published claims and comply with field standards.

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
