# Peer review of "Generation of Primordial Germ Cell-like Cells from iPSCs Derived from Turner Syndrome Patients"

_cells, 2021, doi:10.3390/cells10113099_

Round 1

Reviewer 1 Report

The manuscript entitled " Generation of primordial germ cell-like cells from iPSCs derived from Turner syndrome patients”, by de Souza et al. describes the generation of five human induced pluripotent stem cell lines (hiPSC) from PBMCs of Turner syndrome patients and their differentiation into primordial germ cell-like cell (PGCLCs). By recapitulating the initial germline development, these TS iPSC lines should represent valuable cellular tools to decipher and better understand mechanisms involved in premature primary gonadal failure leading to infertility of TS patients.

Minor comments:
1. Line 376-381: This point is the most important since all further studies will depend of the quality of the TS iPSC lines generated. IPSCs must highly express pluripotency markers. Usually, most of cells (>80-90%) should be double positive for pluripotency markers. All TS lines generated here (and more particularly Pt.2, Pt.4 and Pt.5) do not express sufficiently these markers.

Moreover, flow cytometry results are confusing in histograms. It will be better to replace it by classical analysis with gating to better see scattering and double positive cells.

2. Line 56-65: A paragraph introducing the different strategies used to generate iPSC line (integrative vs integration-free) is mandatory as the paper describes the establishment of TS iPSC lines. However, it would be less confusing if this part had been more synthetic (the paper is not a review). Moreover, use of retroviruses and lentiviruses were the first conventional methods but it is not the case anymore. Since few years most of papers use non-integrative methods (Sendai virus / episomal plasmids) for the establishment of new iPSC lines.

3. Line 72: Please add a space before reference [26-31].

4. Line 88: Please correct by « non-integrating vectors »

5. Line 351: « Table A1»; line 368: « Figure A2»; line 375, … : « Figure A3»: It is the appropriate nomenclature for supplementary table and figures?

6. Line 394: There is no legend for Figure 1E.

7. Figure 2: There are 2 markers of mesoderm and ectoderm germ layers and only one for endoderm germ layer. Please add one marker of endoderm (SOX17 ? as it is used for PGCLC differentiation).

8. Figure 6: Please replace “a” and “b” by classical significance lines.

9. Line 550: Please write “in vitro” in italic.

Reviewer 2 Report

Turner syndrome (TS) is a genetic disorder in females with X Chromosome monosomy associated with highly variable clinical features, which has a huge impact on the physical and mental health of patients. Here, the author demonstrated a robust and efficient method to reprogramming PBMCs-TS into iPSCs, and their differentiation into PGCLCs. Among them, all hiPSCs-TS lines exhibited a pluripotent state capable of differentiating into three germ layers, and PGCLCs-TS summarized the initial germline developmental stages of transcripts and protein markers, including the epigenetic profile. Therefore, these results will facilitate understand the biological process involved in the pathogenesis of TS. However, below are a few comments that are suggested for consideration during its revision.

  1. “These results suggested that PBMCs cell expansion and the number of cells transfected can influence the reprogramming process.” The sentence in the first paragraph is mentioned that the number of transfected cells will affect the reprogramming process, but there is no supportive data.
  2. The third paragraph: In Figure 1D, the expression of KLF4 of hiPSCs-TS PL5 is much higher than that of the hiPSCs control cell line, which is inconsistent with the expression. The fourth paragraph: In Figure 1E, the proportion of hiPSCs-TS PL2 and hiPSCs-TS PL5 expressing NANOG is significantly lower than 70%, which does not match the description.
  3. In the penultimate paragraph, male and female samples were suddenly used as controls, but without any explanation, it was a little bit difficult to understand. It is recommended to explain or reverse the order of the last two paragraphs of the "Generation and characterization of hiPSCs-TS" part.
